# Psychological Aspects of Media Communication during the COVID-19 Pandemic: Insights from Healthcare and Pharmacy Specialists in Lithuania

**DOI:** 10.3390/healthcare9101297

**Published:** 2021-09-29

**Authors:** Nida Žemaitienė, Milda Kukulskienė, Vilma Miglinė, Loreta Kubilienė, Gabrielė Urbonaitė, Laura Digrytė-Šertvytienė, Aušra Norė, Kastytis Šmigelskas

**Affiliations:** 1Faculty of Public Health, Lithuanian University of Health Sciences, 44307 Kaunas, Lithuania; nida.zemaitiene@lsmuni.lt (N.Ž.); Gabriele.urbonaite95@gmail.com (G.U.); laura.digryte@lsmuni.lt (L.D.-Š.); ausra.nore@lsmuni.lt (A.N.); kastytis.smigelskas@lsmuni.lt (K.Š.); 2Community Well-Being Center, Mykolas Romeris University, 08303 Vilnius, Lithuania; vilma.migline@mruni.eu; 3Department of Drug Technology and Social Pharmacy, Faculty of Pharmacy, Lithuanian University of Health Sciences, 44307 Kaunas, Lithuania; Loreta.kubiliene@lsmuni.lt

**Keywords:** COVID-19, pandemics, media communication, risk communication, healthcare workers, pharmacy specialists, mental health, psychological well-being

## Abstract

In the setting of disasters, people seek information as they hope that knowledge will provide security. This makes the media a critical source of crisis exposure. The aim of the study described in this article was to analyze COVID-19 pandemic-related psychological aspects of media use by healthcare and pharmacy workers in Lithuania and to reveal the subjective effects of media consumption on their psychological well-being. 967 healthcare workers and pharmacists in Lithuania completed an electronic survey, which consisted of questions about the changes in well-being experienced since the beginning of the lockdown and media use in the search for information on COVID-19. It was found that communication might have ambiguous effects on psychological well-being. Excessive, unreliable, misleading, contradictory information and “catastrophizing” were subjectively related to impaired psychological well-being. Objective and reliable, relevant, clear, timely, hopeful and supportive information had a subjective positive effect. Seeking COVID-19-related information many times a day was associated with increased fear of becoming infected with COVID-19, feeling unable to control the risk of contracting COVID-19, fear of infecting relatives with COVID-19 and feeling that other people would avoid interaction with healthcare workers and pharmacists because of their job. General browsing was not consistently associated with COVID-19-related fears.

## 1. Introduction

In the setting of natural disasters, such as COVID-19 pandemic, people are more likely to seek information [1], as dependency on the media is heightened during a period of societal change or conflict [2]. This makes the media a critical source of crisis exposure. People hope that knowledge will provide security and help orient themselves in new and unfamiliar situations [2]. In such cases, the media provides an irreplaceable opportunity to inform the public immediately, while at the same time it can become an extremely powerful source of stress, that leads to a significant deterioration in the overall public mental health [3]. Paradoxically, efforts to communicate critical information regarding risks and safety during COVID-19 pandemic, may have a negative impact on the well-being, resulting from the repeated media exposure to the outbreak [4].

It is known that stressogenic information in the media tends to have stressogenic effects [5,6]. Yet in an uncertain critical situation, when a lack of clarity becomes universal, stressogenic information might even have a stronger effect on the well-being [3,7]. In 2020, the top six COVID-19-related terms searched in Google were “coronavirus”, “corona”, “COVID”, “virus”, “corona virus”, and “COVID-19”; at the same time there was a wide spread of false news and conspiracies [8]. Some social media circulated faulty messages, which caused psychological distress both directly and indirectly [9]. World Health Organization (WHO) has warned the public of an “infodemic”—overabundance of excess information [10]. Information overflow may induce panic in affected populations and may create a vicious circle by further exacerbating the stress [11]. The response to outbreaks such as COVID-19 is largely influenced by anxiety over health [12]. Research has shown that information on the number of deaths increases public awareness of the level of threat and reinforces feelings of anxiety [4,13]. Conflicting messages, inaccurate and delayed information may even enhance the harm [14]. It should also be borne in mind that when people are scared or upset, they find it more difficult to understand the information they receive; and when they receive it, they rely on negative information more than positive one [15].

High-standard risk communication helps avoid “infodemics” [16]. It has been emphasized that risk communication messages should be simple, problem-solving and in line with cultural values and background [17]. The speed and accuracy of the first message are of great importance. The first message is best absorbed by readers and is the one that attracts the most attention. All the following information tends to be compared against the first message [18]. Immediate information reduces the sense of uncertainty and greatly helps to understand what happened. This “sense of the event” may prevent one from engaging in harm-inducing activities [18]. Therefore, it is very important to base risk communication strategies on risk perception, societal and cultural factors, and risk attitudes in affected populations [19,20].

It has been shown that COVID-19 pandemic is a very important direct stressor with a great impact on mental health [21]. Frontline healthcare workers (HCWs) and pharmacy specialists are the ones who encounter unprecedented mental distress. The current pandemic of COVID-19 has been reported to be associated with anxiety, stress and depressive symptoms for HCWs [22]. Healthcare workers’ mental health risk and resilience factors have been addressed in recent studies, and clear communication was identified as one of the significant protective factors [23]. Timely information and feedback may act as a buffer against the mental stress of healthcare workers [21]. On the contrary, lack of access to up-to-date information and communication has been pointed out as an important source of anxiety [24]. However, there is a lack of more detailed data on psychological aspects of media communication and subjective effects on the well-being of frontline workers. Thus, it is an issue of great importance to address the needs of those at the core of the COVID-19 pandemic [25]. There is an urgent need for evidence in the field of mental health of healthcare and pharmacy workers, to formulate response-enhancing recommendations [3,26].

Currently, it is presumed that exposure to information can be controlled by controlling the frequency and the content of the information seeking behaviors, and the negative effects on psychological well-being might be reduced during exposure control. However, each new crisis has its own unique characteristics, which might differ, depending on the socio-cultural background. We hypothesized that the more frequent search for pandemic-related information in the media is associated with increased tension and fear in the sample of Lithuanian healthcare and pharmacy workers. This study aims to analyze COVID-19 pandemic-related psychological aspects of media use by healthcare workers and pharmacy specialists, and to reveal the effects of media consumption on their psychological well-being.

## 2. Materials and Methods

A mixed-methods research strategy has been utilized in the present study. The study involved anonymous responses to an electronic survey. Data collection was initiated on 17 August and was completed on 15 October 2020. The electronic survey has been uploaded online for 2 months. The potential participants were invited using institutional emails. The Invitation was sent to 222 target healthcare institutions and pharmacy chains. There were two reminder messages for healthcare professionals’ community to participate in the study. Workers from 56 institutions responded to an invitation and participated in the study. The invitation was additionally publicized in the target social networking groups, training and conferences.

The study involved 967 healthcare workers and pharmacists in Lithuania who completed an electronic survey. Participants of the study were 857 women (88.6%), 101 men (10.4%) and 9 individuals who did not indicate gender (0.9%). 253 of the participants were nurses (26.2%), 252 were physicians (26.2%), 245 were pharmacists (25.3%), 73 were administrative specialists (7.5%) and 130 were other employees (hospital staff, psychologists, etc.) (13.4%) (Table 1). The age of the participants ranged from 21 to 79 years (mean 42.8 years).

Study participants completed an electronic questionnaire on the well-being and information-seeking behaviors during the first wave of COVID-19 lockdown in Lithuania, which lasted from March to June 2020. The questionnaire consisted of 130 statements about study participants’ well-being and experienced changes in physical and psychological well-being, fears, anxiety, stress coping and psychosomatic symptoms. It was not previously used in any study, but was constructed specifically based on the recent research about COVID-19 issues. The questionnaire was tested in a small-scale pilot study. For further analysis, data on COVID-19-related fears and information-seeking behaviors and perceptions were analyzed. The items were created specifically for this study and did not comprise scales with scoring.

COVID-19-related fears were estimated using five items on fears: (1) to infect oneself; (2) to lose control over the infection; (3) to die from this disease; (4) to infect relatives; and (5) to feel anxious about other people avoiding contact with one’s family. The responses were “yes”, “partly” or “no”.

Information-seeking behaviors and perceptions were assessed using three items: (1) seeking in media for the COVID-19-related information during the lockdown (four response options from “never” to “many times a day”); (2) how the information affected you? (open-ended item); (3) general browsing online for various reasons (five response options from “never” to “very often”).

An open-ended question was “How did the information in the media most often affect you?” This item was responded to by 795 participants. In this regard, the goal was to reveal the subjective media communication effects on the responder’s well-being and based on personal experiences of the workers to provide practice-oriented guidelines for the media communication.

The quantitative data analysis was performed using the “IBM SPSS Statistics for Windows, Version 25.0, Armonk, NY, USA: IBM Corp.” software. A univariate analysis included the calculation of prevalence (n and percentages) and means with standard deviations. Data that were missing was not included, because it comprised less than 10% of the responses and was inconsistent. The main analytical perspective was non-parametric, because several essential variables were distributed non-normally (based on skewness and kurtosis). The inferential statistic was conducted using a binary logistic regression in univariate (crude) and multivariate (adjusted) modelling. In the analyses seeking the COVID-19-related information in the media, responses of “never” were not included in the analyses due to a small number of cases in this response group. The statistical significance was set at *p* < 0.05.

The qualitative data analysis was performed using the “MAXQDA Analytics Pro, 2000” software. The thematic analysis method has been applied [27]. A total of 795 answers to an open-ended question were inductively coded. Qualitative coding was performed by one encoder, specializing in inductive qualitative psychology. The primary coding scheme and significant statements that make up each sub-theme were validated by the expert group. Then the codes were read, checked and corrected again. Changes to the structure and sub-themes formulation were made based on the expert’s feedback. There were 1180 codes that were assigned to the language of the participants. The codes were categorized into 21 code categories, five sub-themes and three main themes. Based on these findings and the participants’ quotations, theoretical models about subjective negative and positive media communication effects were built.

## 3. Results

### 3.1. Information Seeking, Browsing Online and Fear Related to the COVID-19 Pandemic

Overall, 76.1% of participants reported having been seeking COVID-19-related information during the lockdown in the media every day (43.1% once-twice a day, 33% many times a day).

Fear of infection of COVID-19 during the lockdown was reported by 41.9% of participants, and 43.8% of participants partly agreed that they were afraid of being infected. Feeling unable to control the risk of infection of COVID-19 was reported by 27.1%, and partly agreed was reported by 46.2%. The majority (65.9%) of participants agreed that they felt the fear of infecting relatives with COVID-19. Feeling anxious that others would avoid interaction with them because of their job was reported by 27.9%, partly agree was reported by 30.1%, and the majority (41.9%) disagreed.

In the total sample, the majority (68.4%) of participants were browsing online during the lockdown frequently. The percentage distribution of answers is shown in Table 2.

### 3.2. Predictors of Fear Related to the COVID-19 Pandemic

In the univariate analysis, increasing age, occupation and seeking COVID-19-related information in media during the lockdown were found to be the strongest significant predictors for different types of fear related to the COVID-19 pandemic.

While adjusting for gender, age, occupation and browsing online, several findings emerged. Seeking COVID-19-related information in the media many times a day was associated with increased fear of becoming infected with COVID-19 during the pandemic (OR = 2.68), feeling unable to control the risk of COVID-19 (OR = 1.70), fear of infecting relatives with COVID-19 (OR = 2.76) and feeling that other people would avoid interaction with them because of their job (OR = 1.90) (all *p* < 0.05). In addition to that, a less frequent searching for COVID-19-related information (once or twice a day) was associated with a significantly increased risk of fear of becoming infected with COVID-19 (OR = 1.50), and fear of infecting relatives (OR = 1.63). Fear of death related to COVID-19 was not statistically significantly associated with COVID-19 related information seeking (Table 3). It should also be noted that casual, not necessarily pandemic-related browsing was consistently not associated with any type of COVID-19-related fears (*p* > 0.05). Values in the bold font indicate statistically significant results. *p* values are presented, when OR is less than 1, or greater than 1.

### 3.3. Subjective Effects of Media on Well-Being

During the qualitative analysis three main themes were identified (the number of codes assigned to a theme is indicated in parenthesis): 1. Negative Effects of Media Communication (874); 2. Positive Effects of Media Communication (176); 3. Neutral or Ambiguous Effects of Media Communication (130). Each theme, its sub-themes and code categories are described below. The results are illustrated using anonymous quotations.

#### 3.3.1. Negative Effects of Media on Well-Being

The first theme consisted of three sub-themes defining the subjective negative effect of media communication: 1.1. Negative Aspects of Communication (110); 1.2. Subjective Negative Impact (711); 1.3. Strategies of Personal Media Usage Management (53). The model below represents subjective relationships of sub-themes: participants have mentioned certain negative aspects of Lithuanian media communication during the first COVID-19 lockdown, that were subjectively associated with the negative impact on psychological well-being. This subjective negative impact was also related to the personal media usage strategies that were used by study participants, that subjectively moderated the negative impact (Figure 1). Numbers in parenthesis indicate how many responses were in each category.

The first sub-theme, Negative Aspects of Media Communication (110), was described by four code categories. Excessive Information (46) had a subjective negative effect for some participants. It was characterized as an incomprehensible flow of information, overly negative and repetitive information. Participants defined it as one-sided, “inflated”, and making them feel in constant exposure. Unreliable and Misleading Information (39) was described as false, misleading, chaotic and inaccurate information, which was characterized by a lack of journalistic or political competence. Contradictory Information (13) was also assessed especially negatively by some of the participants. Conflicting messages and inconsistencies between theory and practice caused confusion. Furthermore, “catastrophizing” (12), hyperbolizing and exaggeration of the negative consequences of the pandemic induced panic and helplessness.

The second sub-theme, Subjective Negative Impact (711), consisted of six code categories describing negative changes in well-being in relation to media usage. The code category General Undifferentiated Negative Effect (241) consisted of short general responses about subjective effects of media communication to healthcare workers and pharmacy specialists; this included such answers as “negative”, “negatively”, “poor”, and “bad.” One of the most saturated categories was Increased Anxiety and Tension (204). It was revealed that in response to information in the media, participants felt stressed and anxious, and also described general tension in their bodies. Specialists also named the reactions of fear, intimidation and panic (146) associated with the media exposure. Participants were frightened by intimidating messages in the media and reactions in society. Irritability and anger (65), a sense of insecurity and loss of control (33), and gloominess and fatigue (22) were also mentioned in association with subjective media effect.

The third sub-theme Strategies of Personal Media Usage (53) included two code categories. Compulsive Browsing (7) was defined as a personal over-involvement in the media, and frustration with browsing for information about the COVID-19 pandemic. Participants described that they could not stop searching for the information. This process was very engaging and almost addictive. Even though this was how they hoped to feel safer, such compulsive browsing considerably increased their anxiety and panic. Another strategy chosen by the participants was Keeping the Distance (46) from overexposure in the media. Some people revealed that they needed to take a break from the media to distance themselves from emotional information, and that this information was tiring and stressful. In Table 4, each category is illustrated with a quote.

#### 3.3.2. Positive Effects of Media on Well-Being

The second theme consisted of two sub-themes, which define the subjective positive effect of media communication: 1.1. Positive Aspects of Communication (109); 1.2. Subjective Positive Impact (67). The model below represents subjective relationships of sub-themes: Participants have mentioned positive aspects of Lithuanian media communication during the first COVID-19 lockdown, that were subjectively associated with the positive impact on psychological well-being (Figure 2).

The sub-theme Positive Aspects of Communication (109) consisted of four code categories. The study participants revealed the necessity of Objective and Reliable Information (81) in the media. Subjective satisfaction with media communication was increased by relying on trustworthy sources and evidence-based information. By selecting reliable sources of information, controlling time on social networks or completely refusing to take an interest in media communication, they sought to reduce the intense negative impact on their emotional well-being. The Code category Relevant Information (19) reveals that media communication should reflect the needs of a reader, and satisfy their interest and curiosity. Participants also mentioned that it was valuable to receive clear and timely information (5), which is not delayed and is provided in a consistent and comprehensible manner. The Code category Hopeful and supportive information (4) defines the need for hopeful, positive and empowering messages in the media.

The second sub-theme, Subjective Positive Impact (67), consisted of five code categories, which defined subjective positive changes in well-being related to the media usage. General Undifferentiated Positive Effect (7) consisted of short responses that the media affected participants positively or had a good effect in general. The Soothing Effect (27) was described as a relaxing and calming impact of media consumption. Participants felt calmer, with a clear understanding of the current situation. This helped reduce their feeling of uncertainty. The Category Compliance with Recommendations (15) revealed that proper communication provided clear guidelines and encouraged adherence to recommendations for safe behavior and hygiene. A positive impact of media consumption on participant’s well-being in several cases also revealed a Sense of Control and Safety (13) and Better Adaptation (4) to the pandemic-related changes. In Table 5, each category is illustrated with a quote.

#### 3.3.3. Neutral or Ambiguous Effects of Media Communication

The last theme included brief answers about the neutral impact of the media. Some of the participants stated that they did not use the media or did use it but did not feel any impact on their well-being: “[The media] didn’t make a difference” (S9). There were also several undifferentiated responses to this theme that the media impact was unrefined and ambiguous: “It was varied. Depended on what content was presented and how” (S29).

## 4. Discussion

In various health-related emergencies, the public relies on the media to obtain factual and timely information to make informed decisions about safe behavior. In the context of an uncertain situation, people start to rely more on the information provided by the media [2]. This is of utmost importance for civil society and healthcare workers as well. Our study shows several factors of media communication affecting healthcare workers well-being in the first wave of the COVID-19 pandemic. During the outbreak, most participants reported seeking COVID-19-related information during the lockdown in the media every day (45.1% once-twice a day, 33% many times a day).

Our study showed that the media became an important factor that influenced psychological well-being. We observed that seeking COVID-19-related information in the media many times a day was associated with increased fear of becoming infected with COVID-19, feeling unable to control the risk of this infection, fear of infecting relatives with COVID-19 and feeling that other people would avoid interacting with them because of their job. In addition, a less frequent searching for COVID-19-related information (once or twice a day) was associated with a significantly increased risk of fear of becoming infected with COVID-19, and fear of infecting relatives. Fear of death related to COVID-19 was not statistically significantly associated with COVID-19 related information seeking. Furthermore, our results show that casual pandemic-related browsing was not associated with any type of COVID-19-related fears. These findings remained significant after adjustment for gender, age and occupation. This is in accordance with another recent study where daily media use has been found to be a significant predictor of stress and depressive symptoms [11,13]. A study in China found that healthcare workers who had been excessively exposed to COVID-19-related information showed a higher incidence of depression, anxiety and insomnia [28]. Mass media coverage of a pandemic can potentially lead to mass hysteria and fear [29]. This has been observed over the 2005 avian flu outbreak. One of the studies indicated that more intense television exposure has been associated with increased fear of the avian flu [30]. Thus, this may explain why the higher flow of information on COVID-19 from various sources predicted greater COVID-19-related fears in our study participants.

It has also been observed that females were more likely than males to search for COVID-19 related information. Similar findings have been reported by a group of researchers in Taiwan in their cross-sectional survey of more than 2000 participants [31]. It is noticed, that there is a tendency for women to search for health-related information online more often [32]. The authors stated that this might reflect their self-esteem and informational support needs. According to the researchers, an online search for men is usually determined solely by informational intentions [32]. Overall, it has been suggested that females are generally more concerned about health-related information than males [31]. Interestingly, COVID-19 related information-seeking behavior was more characteristic of healthcare workers and the related administrative sector. This can be explained by an effort to reshape the adaptation to turbulence and improve the problem-solving skills of administrative personnel [33].

A qualitative thematic analysis of participants’ responses to the open-ended survey question has revealed that certain negative aspects of communication, such as excessive, unreliable and misleading, contradictory information in the media and “catastrophizing”, were subjectively related to impaired psychological well-being. Some of the participants described that media communication had a general undifferentiated negative effect, was associated with increased anxiety and tension, fear, intimidation and panic, caused irritability and anger, and several participants’ responses were linked to increased insecurity, loss of control or gloominess, and fatigue. Comparing this data to the recent survey of Iranian adults during the current pandemic had similar findings, where the more intense media use was associated with psychological distress, fear of COVID-19, misunderstanding of COVID-19 (incorrect information, misconceptions and falsehoods) and insomnia [9].

In our study certain positive aspects of communication, such as objective and reliable, relevant, clear, timely, hopeful and supportive information, were subjectively linked with the general undifferentiated positive, soothing effect on well-being, encouraged compliance with recommendations, was subjectively linked with an increased sense of control, safety or better adaptation to the pandemic situation in several cases. In a similar line, it has been shown that proper use of social media may be a very effective way to deliver timely and crucial information [34]. Accordingly, previous studies showed that exact and detailed information is associated with a lower negative impact on mental health [35].

It was revealed that the negative subjective impact of the media on well-being was moderated by the chosen strategy of personal media usage. Compulsive browsing had a subjective negative impact for several participants of the study and the necessity to keep the distance from the informational overload was emphasized. Similarly, a multinational survey in India, Malaysia, Mexico and the UK showed that compulsive internet and social media use has been strongly related to depression, loneliness and escapism [36].

### 4.1. Strengths and Limitations of the Study

Our study provides additional insight into the COVID-19 pandemic-related tendencies in media use by healthcare and pharmacy specialists and seeks to reveal the subjective negative and positive effects of media use on psychological well-being.

There are some limitations in the present study. This study is specific to Lithuanian society. It is limited by using a single measurement of COVID-19 related media use and its effects in one data collection period: summer 2020. We must admit it is not clear how, where and what information about COVID-19 was sought by the study participants. Longitudinal data were not collected in this investigation as well. In addition, the study does not reflect data on changes in financial situation, lifestyle and working conditions. Finally, all data were self-reported, and the bias in collecting this type of data cannot be avoided. For example, participants may be prone to provide socially desirable responses.

The strengths of the study are a high response rate, the nationally representative data of Lithuanian healthcare workers and pharmacy specialists, and data triangulation, which was ensured by applying a mixed study design. This study contributes to the scientific knowledge base, regarding psychological aspects of media communication and healthcare and pharmacy workers’ mental health. The results are specific to the context of the unprecedented and long lasting COVID-19 pandemic, when faced with an increased media exposure. This study provides knowledge for further analysis of the development of communication strategies in critical situations. This may help to prepare for future crises in a more structured and appropriate way.

### 4.2. Practical Recommendations and Insights for Future Research Directions

At the practical level, healthcare and pharmacy workers should pay attention to their habits of media consumption and properly assess the level of one’s exposure to information. It is recommended to read news about COVID-19 once a day or less frequently. It is also advisable to reduce browsing time, choose reliable sources of information, rely on objective facts instead of emotional information, critically evaluate the content and choose healthier coping and relaxation strategies, when possible. Furthermore, it is advisable to avoid compulsive browsing, focusing on the headlines and constantly thinking about the pandemic.

Appropriate media communication is essential to reduce public anxiety, tension, fear, panic and enhance a positive response to the effects of the pandemic. A calmer public response, adherence to recommendations, a sense of self-control and security are related to certain aspects of media communication. It is strongly recommended that information be presented based on facts by avoiding emotional and “catastrophizing” content. The information must be reliable, non-misleading and non-contradictory. The information should be provided on time, but it is very important to avoid any overflow. Lastly, many issues need to be clarified concerning effective communication on the risks of a public health crisis, in particular the use of the media and its impact. Undoubtedly, scientists will carry out more research on the recent COVID-19 outbreak. This will contribute to the knowledge that can be relied on in the field of public health now and in the future.

## 5. Conclusions

Seeking COVID-19-related information in the media many times a day was associated with increased fear of becoming infected with COVID-19 during the pandemic, feeling unable to control the risk of COVID-19, fear of infecting relatives with COVID-19 and feeling that other people would avoid interacting with them because of their job.General and casual, not necessarily pandemic-related browsing was consistently not associated with any type of COVID-19-related fears.Excessive, unreliable and misleading, contradictory information in the media and “catastrophizing” were subjectively related to impaired healthcare workers’ and pharmacists’ psychological well-being (and an increased tension and anxiety, fear, irritability, insecurity, gloominess and fatigue).Objective and reliable, relevant, clear, timely, hopeful and supportive information in the media had a subjective positive effect (had a soothing effect, encouraged compliance with recommendations, increased sense of control and better adaptation).

## Figures and Tables

**Figure 1 healthcare-09-01297-f001:**
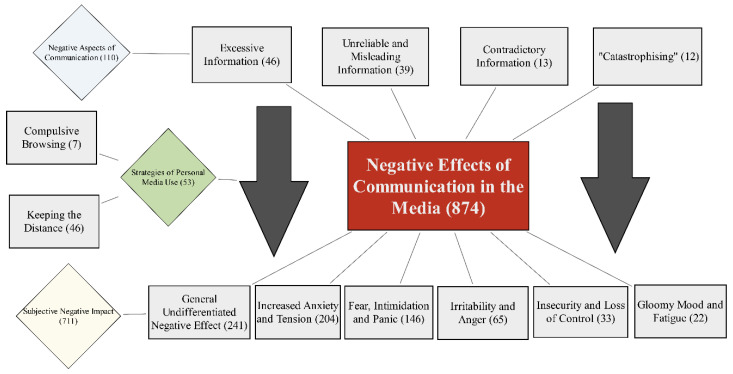
Thematic Model: Negative Effects of Communication in the Media.

**Figure 2 healthcare-09-01297-f002:**
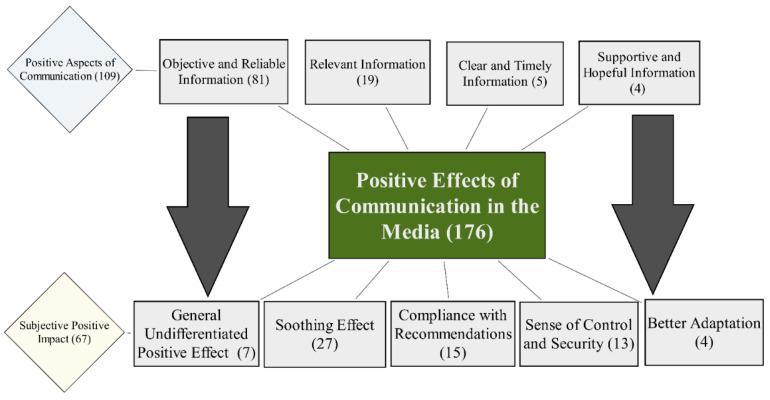
Thematic Model: “Positive Effects of Communication in the Media”.

**Table 1 healthcare-09-01297-t001:** Basic Characteristics of the Study Sample. Baseline Characteristic.

Variable		*n*	%
Gender	women	857	88.6
	men	101	10.4
Age	years, mean ± SD	42.75 ± 12.7	
Work field	public	669	69.2
	private	267	27.6
Healthcare level	primary	172	17.8
	secondary	200	20.7
	tertiary	325	33.6
	none	206	21.3
Profession	physicians	252	26.1
	nurses	253	26.2
	pharmacists	245	25.3
	administrative staff	73	7.5
	others	130	13.4

**Table 2 healthcare-09-01297-t002:** COVID-19-Related Information Seeking, Fears and General Browsing: Percentage Distribution.

Variable	Meaning	*n*	%
Seeking COVID-19-related information in media during the lockdown	never	21	2.2
	several times a week	184	19.7
	once-twice a day	421	45.1
	many times a day	308	33
Fear of getting infected with COVID-19 during lockdown	yes	375	41.9
	partly	392	43.8
	no	127	14.2
Fear of death related to COVID-19	yes	75	8.8
	partly	206	24.2
	no	570	67
Feeling unable to control the risk of infection of COVID-19	yes	234	27.1
	partly	399	46.2
	no	231	26.7
Fear of infecting relatives with COVID-19	yes	592	65.9
	partly	225	25.1
	no	81	9
Feeling anxious that people would avoid interaction with me because of my job	yes	244	27.9
	partly	263	30.1
	no	366	41.9
Browsing online	never	27	3
	rarely	70	7.8
	sometimes	188	20.8
	frequently	306	33.9
	very frequently	312	34.6
Browsing online (two categories)	rarely	285	31.6
	frequently	618	68.4

**Table 3 healthcare-09-01297-t003:** Predicting Factors for COVID-19-Related Fears *.

Variable		Fear of Getting Infected with COVID-19 during Lockdown		Fear of Death Related to COVID-19		Feeling Unable to Control the Risk of COVID-19 Infection		Fear of Infecting Relatives with COVID-19		Feeling Anxious that Others Would Avoid Interaction with Them Because of Their Job	
		OR	*p*	OR	*p*	OR	*p*	OR	*p*	OR	*p*
Seeking COVID-19 related information in the media	several times a week	1.00		1.00		1.00		1.00		1.00	
	once-twice a day	**1.50**	0.044	1.23	0.605	0.89	0.600	**1.63**	0.015	1.09	0.716
	many times a day	**2.68**	0.000	1.93	0.110	**1.70**	0.027	**2.76**	0.000	**1.90**	0.007
Gender	men	1.00		1.00		1.00		1.00		1.00	
	women	1.21	0.430	1.02	0.951	1.63	0.087	1.13	0.623	0.86	0.585
Age		0.99	0.212	**1.03**	0.008	1.00	0.710	**0.96**	0.000	**0.97**	0.000
Profession	physicians	1.00		1.00		1.00		1.00		1.00	
	nurses	1.09	0.682	**0.47**	0.046	0.74	0.212	1.42	0.113	**1.57**	0.049
	pharmacists	**1.68**	0.009	0.80	0.527	**2.22**	0.000	**1.64**	0.029	**2.03**	0.001
	administrative staff	**0.51**	0.037	0.64	0.397	0.48	0.068	**0.39**	0.003	0.76	0.483
	others	0.72	0.181	0.63	0.306	0.74	0.301	0.83	0.455	0.87	0.617
Browsing online	rare	1.00		1.00		1.00		1.00		1.00	
	frequent	1.04	0.561	1.12	0.415	1.04	0.663	1.06	0.421	1.02	0.810

* Values in the bold font indicate statistically significant results.

**Table 4 healthcare-09-01297-t004:** Theme: Negative Effects of Communication in the Media: Sub-themes, Categories and Illustrative Quotes.

Sub-Theme	Category	Illustrative Quote (Segment No.)
Negative Aspects of Communication (110)	Excessive Information (46)	“There was the only talk of a virus as if nothing more happened in the world. It is a pity that there was no positive information. I really believe that there were a lot of interesting things going on at the time which weren’t presented to the public.” (S23)
	Unreliable and Misleading Information (39)	“The incompetence of journalists in medical matters has a negative effect.” (S147)
	Contradictory Information (13)	“The information was contradictory, inaccurate.” (S246)
	“Catastrophizing” (12)	“It was determined to wait for the worst.” (S373)
Subjective Negative Impact (711)	General Undifferentiated Negative Effect (241)	“Negative”, “negatively”, “bad.”
	Increased Anxiety and Tension (204)	“Only negative messages gave cause for concern.” (S13)
	Fear, Intimidation and Panic (146)	“Information was more intimidating than reassuring” (S754); “It always caused unnecessary panic and negative emotions.” (S654)
	Irritability and Anger (65)	“This [information] annoyed me.” (S397)
	Insecurity and Loss of Control (33)	“It was not clear how to proceed, there was no protective equipment.” (S779)
	Gloomy Mood and Fatigue (22)	“The news was depressing, especially death reports” (S209); “Usually it is tiring.” (S255)
Strategies of Personal Media Usage (53)	Compulsive Browsing (7)	“I couldn’t back down because I was afraid to miss something.” (S70)
	Keeping the Distance (46)	“The headlines sent a very bad message, I only tried to rely on the facts.” (S53)

**Table 5 healthcare-09-01297-t005:** Theme: Positive Effects of Communication in the Media: Sub-themes, Categories and Illustrative Quotes.

Sub-Theme	Category	Illustrative Quote (Segment No.)
Positive Aspects of Communication (109)	Objective and Reliable Information (81)	“The more information about medical experience, the better. You need to know what’s going on in the world” (S147); “I was just watching the news and I was only reading official information—the situation seemed to be under control.” (S19)
	Relevant Information (19)	“It was interesting to follow the situation in Lithuania and around the world.” (S116)
	Clear and Timely Information (5)	“I felt reassured by the daily info provided by the manager of emergency situations.” (S176)
	Hopeful and Supportive Information (4)	“Partly encouraged me to feel more positive (seeing physicians being supported and encouraged to work).” (S217)
Subjective Positive Impact (67)	General Undifferentiated Positive Effect (7)	“Positively.”
	Soothing Effect (27)	“Sometimes it reassured me.” (S195)
	Compliance with Recommendations (15)	“Initially, it allowed me to understand what is happening, what requirements are applied, how to proceed.” (S162)
	Sense of Control and Safety (13)	“It helped me control the situation.” (S5)
	Better Adaptation (4)	“I mainly read, watch or listen to the news so that I can prepare myself properly.” (S309)

## Data Availability

All databases are available from the corresponding author upon reasonable request.

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
