# Peer review of "Psychological Aspects of Media Communication during the COVID-19 Pandemic: Insights from Healthcare and Pharmacy Specialists in Lithuania"

_healthcare, 2021, doi:10.3390/healthcare9101297_

Round 1

Reviewer 1 Report

This research examined COVID-19 pandemic related tendencies in media use by healthcare workers and pharmacy specialists.

Although the authors have made considerable efforts to develop this paper, I believe that the current version of the manuscript should be improved through significant revision and re-writing. I want to provide some suggestions for the improvement of this paper as follows.

[1] Introduction

I think that the overall structure and writing of the introduction part are not clear and well-aligned because it is not easy to catch the research questions and strategies in this paper. Please clearly describe those things. As you already knew, the introduction section is one of the most important parts to draw readers' attention and provide guidelines for them to facilitate a clear understanding of the paper.

[2] Theories and hypotheses

This paper did not provide the part of “Theory and Hypotheses. So, it is very difficult for me to ensure that the research has enough theoretical value and contribution. I think that this is the critical flaw of this paper. Would you please provide the part in an elaborated way?

- Although this paper dealt with interesting phenomena, it did not provide adequate theoretical background and support for the development of its hypotheses. This is the critical limitation of this paper. 

[3] Strengths and Limitations of the Study

- Although the authors have attempted to explain the contributions and implications of the paper, I think that the overall quality of the explanations is low. Please provide more elaborated explanations to demonstrate its theoretical and practical contributions.

Author Response

Good day,

We are truly grateful for Your valuable comments and insights. Please see the attachment below with our revised and improved version of the manuscript. 

[1] Introduction:

Point 1: I think that the overall structure and writing of the introduction part are not clear and well-aligned because it is not easy to catch the research questions and strategies in this paper. Please clearly describe those things. As you already knew, the introduction section is one of the most important parts to draw readers' attention and provide guidelines for them to facilitate a clear understanding of the paper.

Response 1: The introduction was significantly revised and re-written. The structure of the introduction was revised and changed, in order to convey the research question more consistently and clearly. The aim was to re-write the introductory part so that it would be more appealing to the reader and provide general aspects of the publication.

[2] Theories and hypotheses:

Point 2: This paper did not provide the part of “Theory and Hypotheses. So, it is very difficult for me to ensure that the research has enough theoretical value and contribution. I think that this is the critical flaw of this paper. Would you please provide the part in an elaborated way? Although this paper dealt with interesting phenomena, it did not provide adequate theoretical background and support for the development of its hypotheses. This is the critical limitation of this paper. 

Response 2: Part of the theory and hypotheses was supplemented, incorporating new literature sources and refining the theoretical rationale. This part in a revised publication is provided in an elaborated way.  As a primary idea of this mixed-methods study was to apply an abductive research strategy, a broad and open research question was used. Though, according to the reviewer's suggestions, we have provided more theoretical background. Furthermore, we have refined and clearly formulated our hypotheses.

[3] Strengths and Limitations of the Study:

Point 3: Although the authors have attempted to explain the contributions and implications of the paper, I think that the overall quality of the explanations is low. Please provide more elaborated explanations to demonstrate its theoretical and practical contributions.

Response 3: The section on research contributions was significantly expanded and supplemented. Recommendations for future research were supplemented by practical recommendations for healthcare and pharmaceutical professionals at the individual level.

[English language and style]

Point 4: English language and style are fine/minor spell check required

Response 4: The English language and style were edited by two impartial and unrelated English language professionals (a translator and a teacher).

Reviewer 2 Report

I found this article to be extremely important, timely, relevant, helpful, and well-researched, and I would certainly like to see it published in this journal. However, there are many problems with the English language text, and various errors using APA (American Psychological Association) Publication Manual 7th Edition format, which is what I am using in my feedback below.  However, the journal editors may choose to utilize a different format in this regard which is their decision. In my feedback I am using quotation marks just to indicate what I am focusing on.  As the detailed grammatical revisions I am indicating are extensive, I am just giving some representative examples here and I am requesting that the authors carefully go through their article and make all repeated and/or similar revisions that I have indicated.

Author Response

Good day,

We are truly grateful for Your valuable comments and insights. Please see the attachment below with our extensive corrections made, based on the attachment in Your review.

[English language and style]

Point 1: Extensive editing of English language and style required. <...> However, there are many problems with the English language text <...>

Response 1: The English language and style were edited by two impartial and unrelated English language professionals - a translator of scientific articles and a translator/English teacher.

Point 2. However, there are many problems with <...> various errors using APA (American Psychological Association) Publication Manual 7th Edition format, which is what I am using in my feedback below.  However, the journal editors may choose to utilize a different format in this regard which is their decision. In my feedback I am using quotation marks just to indicate what I am focusing on.  As the detailed grammatical revisions I am indicating are extensive, I am just giving some representative examples here and I am requesting that the authors carefully go through their article and make all repeated and/or similar revisions that I have indicated.

Response 2: We have carefully edited our article, according to Your indicated revisions and examples in a reviews' attachment. All highlighted items were corrected, however, the authors decided to stick to the journal guidelines for the formatting of the References. The References were written on the basis of the following document: https://mdpi-res.com/data/mdpi_references_guide_v5.pdf. This document is provided in a journal's Healthcare official website. Also, the References section has been cross-checked with articles published by other authors in this particular journal.

Reviewer 3 Report

The aim of this study was to analyze COVID-19 pandemic related tendencies in media use by healthcare workers and pharmacy specialists and to reveal the effects of media use on the psychological well-being.

The paper is very interesting. Below are some constructive suggestions to strengthen your research paper. 

Title

I recommend adding "in Lithuania" at the end of the title

Abstract

Please remove ref [1]

Introduction

  1. Please provide reference to the sentence in L 41-43. I recommend: Dopelt K, Bashkin O, Davidovitch N, Asna N. Facing the unknown: Healthcare workers' concerns, experiences, and burnout during the COVID-19 pandemic - a mixed methods study. Sustainability, August 2021; 13(16), 9021-9034; https://doi.org/10.3390/su13169021.
  2. Please add your research hypothesis

Methods

  1. “Study participants completed an electronic questionnaire on the well-being and organizational challenges” – I didn’t see any organizational challenges questionnaire. It also wasn’t mentioned in the aim of the study.
  2. Please explain the questionnaire validation process.
  3. How was the questionnaire distributed? How many entries were in the questionnaire? How long has he been online? Were reminders sent?
  4. Were the variables distributed normally? How was it tested?
  5. Who encodes the themes? One or several encoders? What was the correlation between them?

Results

well-written

Discussion and conclusions

Beyond recommendations for future research, what are the practical recommendations in light of the findings?

Author Response

Good day,

We are truly grateful for Your interest in our study, and, especially, your constructive comments and insights. Please see the attachment below with our revised and improved version of the manuscript.

[Title]

Point 1: I recommend adding "in Lithuania" at the end of the title

Response 1: Added

[Abstract]

Point 2: Please remove ref [1]

Response 2: Removed

[Introduction]

Point 3: Please provide reference to the sentence in L 41-43. I recommend: Dopelt K, Bashkin O, Davidovitch N, Asna N. Facing the unknown: Healthcare workers' concerns, experiences, and burnout during the COVID-19 pandemic - a mixed methods study. Sustainability, August 2021; 13(16), 9021-9034; https://doi.org/10.3390/su13169021.

Response 3: Reference was provided

Point 4: Please add your research hypothesis

Resoponse 4: Research hypothesis was added (line 86-95).

"Currently, it is presumed that exposure to information can be controlled by controlling the frequency and the content of the information-seeking behaviors, and the negative effects on psychological well-being might be reduced during exposure control. However, each new crisis has its own unique characteristics, which might differ, depending on the socio-cultural background. We hypothesized that the more frequent search for pandemic-related information in the media is associated with increased tension and fear in the sample of Lithuanian healthcare and pharmacy workers. This study aims to analyze COVID-19 pandemic-related psychological aspects of media use by healthcare workers and pharmacy specialists, and to reveal the effects of media consumption on their psychological well-being."

[Methods]

Point 5: “Study participants completed an electronic questionnaire on the well-being and organizational challenges” – I didn’t see any organizational challenges questionnaire. It also wasn’t mentioned in the aim of the study.

Response 5: The inaccuracy has been corrected

Point 6: Please explain the questionnaire validation process.

Response 6: The questionnaire validation process was explained in the methodological part: lines 173-175.

"It was not previously used in any study but was constructed specifically based on the recent research about COVID-19 issues. The questionnaire was tested in a small-scale pilot study."

Point 7: How was the questionnaire distributed? How many entries were in the questionnaire? How long has he been online? Were reminders sent?

Reponse 7: Questionnaire distribution processes were explained in a revised publication (lines 150-157)

"Data collection was initiated on August 17 and was completed on October 15, 2020. The electronic survey has been uploaded online for 2 months. The potential participants were invited using institutional emails. The invitation was sent to 222 target healthcare institutions and pharmacy chains. There were two reminder messages for the healthcare professionals’ community to participate in the study. Workers from 56 institutions responded to an invitation and participated in the study. The invitation was additionally publicized in the target social networking groups, training, and conferences."

Point 8: Were the variables distributed normally? How was it tested?

Response 8: The distribution of the variables was explained in a revised publication (lines 198-200).

"The main analytical perspective was non-parametric because several essential variables were distributed non-normally (based on skewness and kurtosis)."

Point 9: Who encodes the themes? One or several encoders? What was the correlation between them?

Response 9: The encoding process was thoroughly explained in a revised publication (lines 208-211).

"Qualitative coding was performed by one encoder, specializing in inductive qualitative psychology. The primary coding scheme and significant statements that make up each sub-theme were validated by the expert group. Then the codes were read, checked, and corrected again. Changes to the structure and sub-themes formulation were made based on the expert’s feedback." 

[Discussion and conclusions]

Point 10: Beyond recommendations for future research, what are the practical recommendations in light of the findings?

Response 10: The part of the Recommendations has been expanded. A new paragraph has been added on recommendations for healthcare and pharmacy workers at the practical individual level.

"At the practical level, healthcare and pharmacy workers should pay attention to their habits of media consumption and properly assess the level of one’s exposure to information. It is recommended to read news about COVID-19 once a day or less frequently. It is also advisable to reduce browsing time, choose reliable sources of information, rely on objective facts instead of emotional information, critically evaluate the content, and choose healthier coping and relaxation strategies, when possible. Furthermore, it is advisable to avoid compulsive browsing, focusing on the headlines, and constantly thinking about the pandemic."

The section on research contributions was also significantly expanded and supplemented.

[English language and style]

Point 11: English language and style are fine/minor spell check required

Response 11: The English language and style were edited by two impartial and unrelated English language professionals (a translator and a teacher).

Round 2

Reviewer 1 Report

The authors successfully responded to the reviewer's comments and adapted the scientific paper in accordance with those comments.

Reviewer 3 Report

Well done! No further comments.